# Differential Physiological Response and Antioxidant Activity Relative to High-Power Micro-Waves Irradiation and Temperature of Tomato Sprouts

**Audrius Radzevičius** [1,*] **, Sandra Sakalauskienė** [1] **, Mindaugas Dagys** [2] **, Rimantas Simniškis** [2] **, Rasa Karklelienė** [1] **, Danguolė Juškevičienė** [1] **, Roma Račkienė** [3] **and Aušra Brazaitytė** [1]

[1] Lithuanian Research Centre for Agriculture and Forestry, Institute of Horticulture, Kaunas St. 30, Babtai, LT-54333 Kaunas, Lithuania; sandrasakaluskien@lsdi.lt (S.S.); rasa.karkleliene@lammc.lt (R.K.); danguolejuskeviciene@lsdi.lt (D.J.); ausrabrazaityte@lsdi.lt (A.B.)

[2] Centre for Physical Science and Technology, A. Goštauto 11, LT-01108 Vilnius, Lithuania; mindaugasdagys@lsdi.lt (M.D.); rimantassimniskis@lsdi.lt (R.S.)

[3] Faculty of Electrical and Electronics Engineering, Kaunas University of Technology, Donelaičio St. 73, LT-44249 Kaunas, Lithuania; romarackiene@lsdi.lt

\* Correspondence: audrius.radzevicius@lammc.lt

**Abstract:** Among the various types of stress, microwaves and temperature can induce major impacts on plant growth. There is information describing the thermal impact of microwaves on living organisms, but it is necessary to segregate the warming effect and direct impact of microwaves irradiation on plants. It was detected that High Power Microwaves (HPM) (9.3 GHz) and elevated temperature exposure upon tomato seeds and sprouts in primary ontogenetic stages showed a slightly incentive effect on plant-growing indicators such as dry mass, fresh mass, plants height, and assimilation area. Such a positive effect on plant growing parameters could be related to saccharides distribution by microwaves in seeds or plants and nutrients mobilization. Moreover, tomato plants (+R) and seeds (R) irradiation significantly reduced the content of non-structural carbohydrates (raphinose, glucose, fructose, and sucrose). Obtained results confirm that a common plant acclimatization response to various environmental elements is the concentration of secondary metabolites and antioxidants.

**Keywords:** antioxidant activities; leaf assimilation; chlorophylls; irradiation; microwaves; response; tomato

## 1. Introduction

Plants and environmental factors grant a considerable interest in the research area. There are several signaling pathways that operate in different parts of plants that tends to respond against external stimulation. Among the different types of environmental factors (temperature, drought, salinity, pH, microwaves, etc.), microwaves and temperature can induce major impacts on plant growth [1–3].

Electromagnetic radiation is called microwave radiation, for which its wavelengths are much greater compared with light wavelengths. Microwaves are radio waves with wavelengths that fluctuate from as short as one millimeter to as long as one meter or, tantamount, with frequencies from 300 MHz (0.3 GHz) up to 300 GHz, including an extremely high frequency (EHF) and ultra-high frequency (UHF) with different sources of various boundaries. Radio frequency (RF) engineering often places the upper boundary around 100 GHz (3 mm) and the lower boundary at 1 GHz (30 cm); meanwhile, microwave includes the entire super high frequency (SHF) band (3 to 30 GHz, or 10 to 1 cm) at minimum at all times [1,4]. Microwaves are an example of nonionizing radiation, which do not contain sufficient energy to change substances chemically using ionization. Generally, the heating effect of microwave irradiation accelerates chemical reactions and the affected

molecule realigns itself with the changing field of the medium continually. This produces the result where electromagnetic energy becomes converted into heat energy. Therefore, it can produce harmful or lethal effects on the cells, without prejudice to the chemical bonds affecting the charged particles resulting in dipolar molecules orientation or involving voltage changes in the cell membrane [5–8].

There is information describing the thermal effect of microwaves on living organisms, but in future experiments, it is necessary to separate the thermal impact and actual impact of microwaves irradiation on plants. Scientists, mainly, had been interested in radiofrequency waves and microwaves exposure to human health risks by assessing electromagnetic pollution, and less information is available about the influence of high power microwaves (HPM) upon plants [9,10]. Previous researchers had the aim to increase the output power to tens or hundreds of gigawatts of newly developed HPM devices, but they were confronted with unexpected difficulties due to physical limitations. Some data can be found about the impact of non-thermal microwaves on seeds and plants in their primary ontogenetic development level when their response to external exposure is more sensitive in comparison with other young biological objects [11,12]. In early 2007 years, experiments were made using different strength electromagnetic fields on carrot seeds, and it was found that direct electromagnetic field exposition of various strengths and time periods of action had a different effect on carrot quality, yield, and seed germination dynamics [13]. Ursache studied the impact of non-thermal microwave (MW) electromagnetic waves power density and radio frequency in vegetal tissues upon assimilatory pigments and reported that the levels of photosynthetic pigments were enlarged in both investigated experiments when exposition time was reduced (1 to 4 h), while extended exposition period (12 h) decreased pigment content. Moreover, MW exposure had a slight stimulatory influence on biomass accumulation [10].

Hamada investigated MW radiation at 10.525 GHz frequency with 2.85 cm wavelengths for 75, 45, and 15 min. on wheat grains. It was mentioned that MW radiation decreased the content of phenolic compounds, nucleic acids, and saccharides but increased the content of amino acid and proteins, and it was also observed that proline synthesis was stimulated by a short exposition period (15 min), whereas the other doses had a negative influence on the total proline amount [7].

It is known that microwaves irradiation can affect the functioning of living organisms but the exact biochemical mechanism is not very well clarified. Previous researchers reported that this mechanism may be determined by the electric field frequency and amplitude and, also in the general case, on tit's efficiency cycle, as well as on the type and features of the exposure biological object. Millimeter-waves are known to be immanent to any living organism because a biological object's cell membrane vibrations lie in the mm-wave range frequencies. However, a positive effect on biological organisms can be initiated by millimeter-wave irradiation, which initiates resonant phenomena in living cells [14,15].

With the increase in population, a prevailing way to safe food is to adopt new technologies that could enrich seed germination rates, plant yield, and crop quality improvement [16].

This study aimed to find out the HPM exposure influence on tomato dry seeds, sprouts and on the amount of photosynthetic pigments.

## 2. Materials and Methods

Tomato seeds and plants were exposed by high power microwaves inside waveguides at the Lithuanian Center of Physical Sciences and Technology. The experimental structure is displayed in Figure 1.

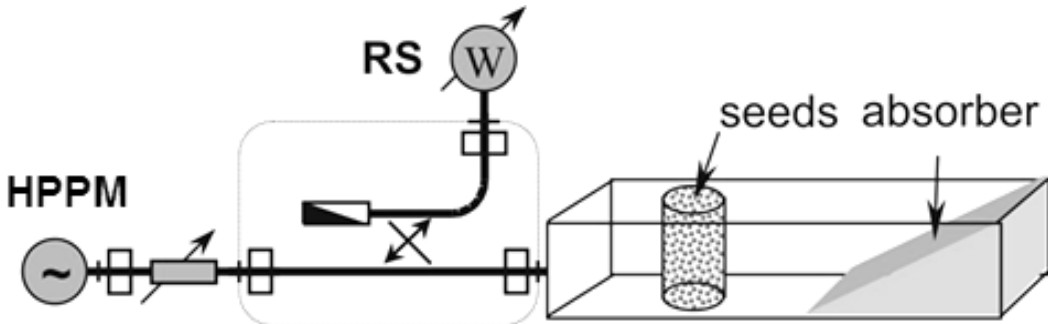

**Figure 1.** Experimental structure: microwave expose on tomato seeds in the waveguide, where RS is a resistive sensor applied as a transmitted microwave power meter; HPPM—high power pulsed magnetron.

In order to maintain a uniformity of electric field distribution inside the waveguide for seeds irradiation, special containers with thin plastic walls were applied. It is supposed that seed humidity is low and the cylindrical container walls are thin. Therefore, the electric field in an empty waveguide as in a container (mounted in the middle of a rectangular waveguide) is the same, and it can be calculated as follows:

$$E = \sqrt{\frac{2z_B P}{ab}};$$

where the following is the case:

$$z_B = \frac{z_0}{\sqrt{1 - \left(\frac{\lambda}{2a}\right)^2}},$$

where $z_B$ is waveguide impedance, $z_0 = 377\ \Omega$ is free space impedance, $\lambda$ is microwave wavelength, and $a$ and $b$ are dimensions of rectangular waveguide.

Investigating the impact of microwave irradiation, the tomato seeds and plants were exposed by HPM. Short high-power rare-repetition microwave pulses were applied to avoid heating thermal seeds by microwave energy.

Seeds and sprouts of tomato variety "Viltis" (Lithuanian, determined type variety) were the study objects. One thousand non-inoculated seeds were selected for the experiment, in which germination energy reached 91.6% and germination reached $-97\%$. The experimental design is displayed in Figure 2.

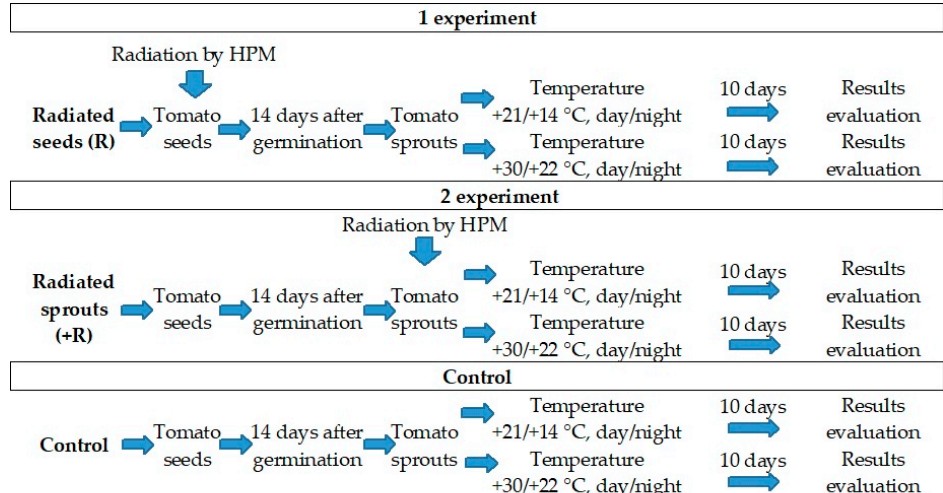

**Figure 2.** Experimental design.

During the first experiment, tomato seeds were irradiated for 10 min using 9.3 GHz frequency microwaves by 4 μs pulse duration and 25 Hz pulse repetition and an electric field at 320 kV m$^{-1}$. At the second experiment, tomato sprouts (14-days old) were irradiated for 20 min using 9.3 GHz frequency microwaves by 4 μs pulse duration and 25 Hz pulse repetition, with an electric field at 320 kV m$^{-1}$ [12]. Temperature exposition started following 14 days after seeds germination. Tomatoes were grown under a controlled environment at the Institute of Horticulture, Lithuanian Research Centre for Agriculture and Forestry (LRCF IH), in a growth chamber under ambient (T—+21/+14 °C, day/night) and elevated (T—+30/+22 °C, day/night) [17] temperatures in pots with peat substrate (produced by Durpeta, Profi 1) (pH 5–6). For artificial illumination, high-pressure sodium lamps (SON-T Agro, Philips) at a PPFD of ~300 μmol m$^{-2}$ s$^{-1}$ were used and the photoperiod reached 16 h. Laboratory analyses were conducted at the Laboratory of Physiology (LRCAF IH) after 24 days from plants germination. The experiment was carried out in three replications, and each replicate consisted of 20 plants.

Carotenoids and chlorophyll determination. The amounts of carotenoid and chlorophyll were determined in green leaves by spectrophotometry in an absolute extract of acetone [18], using a spectrophotometer Genesys 6 (Thermo Spectronic, Madison, WI, USA).

Total phenolic compounds Determination. The content of total phenolic compounds was determined by the calorimetric Folin–Ciocalteau method in extracts of methanol (POCh, Poland) (1 g of plant tissues were grounded and mixed with liquid nitrogen and diluted with 10 mL methanol (80%)). The absorbance was measured at 765 nm after 20 min using spectrophotometer Genesys 6 (Thermospectronic, Thermo Electron Scientific Instruments LLC, Madison, WI, USA) against water as a blank. As the standard used Gallic acid, the total phenolics were established using the calibration method [19].

DPPH determination. The antioxidant activity of plant leaf methanolic extracts (fresh leaf sample 1 g) was evaluated spectrophotometrically via their DPPH-scavenging capacity using spectrophotometer Genesys 6. The solution was shaken for a 30 min period and centrifuged for 20 min at 2012× $g$. After 16 min, the absorbance was recorded at 515 nm and the ability of plant material to scavenge DPPH (μmol g$^{-1}$) was calculated [20].

Ascorbic acid Determination. The amount of ascorbic acid (vitamin C) was determined by a spectrophotometer Genesys 6 (Thermospectronic, USA) [21]. One gram of tomato plant tissues was homogenized in 10 mL oxalic acid (of 5%) (Fluka, Germany) and centrifuged for 5 min, 1691× $g$. Later, the extract (1 mL) was mixed with methyl viologen (2 mL of 0.1%) (Sigma-Aldrich, Germany) and sodium hydroxide (2 mL 2 mol L$^{-1}$) (Delta Chem, Chech rep.) and left to stand for a 2 min time period. The colored radical ion was established at 600 nm compared with the radical blank.

Non-structural carbohydrates determination. High-performance liquid chromatography (HPLC) methods were used to determine fructose, mannose, sucrose, glucose, and raphinose. Fresh plant tissue (about 1 g) was ground and diluted with 4 mL double distilled water (+70 °C). Syringe filters of cellulose acetate (pore diameter 0.25 μm) were used for sample filtering. Shimadzu HPLC (Japan) chromatograph with a refractive index detector (RID 10A) was used for analysis. The oven temperature was maintained at +80 °C. Shodex SC-1011 column (300 × 4.6 mm) (Japan) was used for separation of carbohydrates with a mobile phase using double distilled water.

Biometric measurements. Ten randomly selected plants were measured to assess plant hypocotyl length in centimetres (cm). The fresh mass was measured using calibrated scales by weighing ten plants and expressed in grams (g). The dry mass of tomato sprouts was determined by drying ten tomato plants (each separately) in a drying oven (Czeck Republik, Venticell, MBT) for 24 h at 105 °C [18,22]. The assimilation area was measured for each leaf of every single plant (total number reached 10 sprouts) by scanning with an automatic assimilation (leaf) area meter (England, AT Delta-T Devices, comprises WinDIAS Software).

Statistical analysis. Three analytical replications of measured phytochemicals and ten replications for each treatment of biometric measurements have been executed. For

each analysis, conjugated biological samples of green matter (cotyledons) were used from randomly selected ten tomato plants (0.5–1 g per sample).

All values are expressed as mean ± standard deviation (STDV) and calculated using statistical software Statistica 7 and MS Excel software (version 7.0).

### 3. Results and Discussion

Plant physiological parameters performing the propagation of rapid systemic responses to stress and various environmental stimulations have been the object of investigation, with enlarging arguments targeted to plant photosynthetic pigments, non-structural carbohydrates, and antioxidant properties as a key mediator [23–25]. According to our data, it was observed that elevated temperatures (30 °C) had a reliably positive effect on plant height and assimilation areas compared with tomatoes grown at 21 °C temperature and the control (Table 1). Meanwhile, significantly higher plants and greater amount of fresh mass were established in irradiated tomato (+R) plants grown at 21 °C compared with the control. Tomato (R) plant (from irradiated seeds) growths at 21 °C temperature were significantly higher with a larger assimilation area and higher amount of dry mass compared with no irradiation. The elevated temperature had a positive influence on plant height, assimilation area, and plant fresh mass.

**Table 1.** Growth parameters of tomato cultivated under combinations of different temperature and HPM radiation.

| Impact | Height, cm | STDEV | Dry Weight, g | STDEV | Fresh Weight, g | STDEV | Assimilation Area, cm$^2$ | STDEV |
|---|---|---|---|---|---|---|---|---|
| | | | | T—21 °C | | | | |
| Control | 12.20 | ±0.770 | 0.30 | ±0.017 | 2.81 | ±0.147 | 85.33 | ±5.691 |
| R | 13.30 | ±0.361 | 0.33 | ±0.011 | 2.93 | ±0.078 | 95.33 | ±1.543 |
| +R | 13.47 | ±0.424 | 0.30 | ±0.014 | 3.13 | ±0.094 | 91.70 | ±1.764 |
| | | | | T—30 °C | | | | |
| Control | 18.20 | ±0.529 | 0.27 | ±0.026 | 3.12 | ±0.116 | 74.66 | ±15.823 |
| R | 19.50 | ±0.416 | 0.29 | ±0.019 | 3.52 | ±0.241 | 105.67 | ±8.471 |
| +R | 19.20 | ±0.395 | 0.26 | ±0.030 | 3.20 | ±0.184 | 96.33 | ±5.275 |

Moreover, the higher temperature had a positive effect on the amount of carotenoids, chlorophyll a, and chlorophyll b, but HPM had no any significant changes in the amount of photosynthetic pigments in tomato leaves (Table 2). According to obtained data, we can conclude that the irradiation of HPM and the elevated temperature had a positive effect on tomato sprouts growing parameters such as fresh mass, plants height, and assimilation area. Reports from other researchers include the following: Hamada announced that different doses of microwaves had significantly increased fresh mass, shoots and roots lengths, and succulence in wheat seedlings of 14 days old [7]. Moreover, a positive correlation among succulence and shoot length was admitted. Others reported a positive influence of low intensity electromagnetic microwave field on plant height, growth of buds, and fresh mass, but the stimulus depended on irradiation time and plant species [26]. In a field experiment, the exposition of a pulsed electromagnetic field (PEMF) used for 30 min with 16 Hz frequency had increased germination of soybean seeds up to 8.00% and resulted in a higher yield up to 21% [27].

**Table 2.** Photosynthetic pigment amounts of tomato cultivated under combinations of different temperature and HPM radiation.

| Impact | Carotenoids mg g$^{-1}$, FW | STDEV | Chlorophyll a mg g$^{-1}$, FW | STDEV | Chlorophyll b mg g$^{-1}$, FW | STDEV | Chlorophylls a + b mg g$^{-1}$, FW | STDEV |
|---|---|---|---|---|---|---|---|---|
| | | | | T—21 °C | | | | |
| Control | 0.33 | 0.050 | 1.15 | 0.146 | 0.45 | 0.026 | 1.60 | 0.098 |
| R | 0.35 | 0.012 | 1.21 | 0.091 | 0.45 | 0.020 | 1.66 | 0.080 |
| +R | 0.28 | 0.026 | 1.00 | 0.072 | 0.47 | 0.029 | 1.47 | 0.084 |
| | | | | T—30 °C | | | | |
| Control | 0.41 | 0.030 | 1.51 | 0.121 | 0.68 | 0.067 | 2.19 | 0.203 |
| R | 0.44 | 0.023 | 1.50 | 0.059 | 0.59 | 0.027 | 2.09 | 0.083 |
| +R | 0.46 | 0.019 | 1.43 | 0.083 | 0.69 | 0.059 | 2.11 | 0.061 |

The significantly lower amount of all investigated non-structural carbohydrates (raphinose, sucrose, mannose, fructose, and glucose), which caused a significantly lower total amount of carbohydrates, was determined in tomatoes (+R) and (R) leaves cultivated at 21 °C. Meanwhile, tomato plant (+R) and seed (R) irradiation at higher temperature (30 °C) conditions not only significantly increased the content of mannose but also decreased the amount the other carbohydrates (raphinose, sucrose, fructose, and glucose) compared with the control (Table 3). Such a positive effect on plant growing parameters could be related to saccharides' distribution by microwaves in seeds or plants and nutrients mobilization [28].

**Table 3.** Non-structural carbohydrates of tomato cultivated under combinations of different temperature and HPM radiation.

| Impact | Raphinose, mg g$^{-1}$, FW | STDEV | Sucrose, mg g$^{-1}$, FW | STDEV | Glucose, mg g$^{-1}$, FW | STDEV | Manosse, mg g$^{-1}$, FW | STDEV | Fructose, mg g$^{-1}$, FW | STDEV |
|---|---|---|---|---|---|---|---|---|---|---|
| | | | | | T—21 °C | | | | | |
| Control | 2.02 | 0.261 | 2.53 | 0.315 | 1.69 | 0,475 | 1.98 | 0.510 | 0.66 | 0.028 |
| R | 1.43 | 0.218 | 2.33 | 0.397 | 0.64 | 0.279 | 0.77 | 0.461 | 0.01 | 0.019 |
| +R | 0.87 | 0.447 | 1.08 | 0.283 | 0.47 | 0.237 | 0.28 | 0.297 | 0.23 | 0.370 |
| | | | | | T—30 °C | | | | | |
| Control | 1.48 | 0.139 | 1.23 | 0.147 | 0.68 | 0.230 | 0.00 | 0.000 | 1.21 | 0.405 |
| R | 1.02 | 0.217 | 0.15 | 0.083 | 0.04 | 0.015 | 0.37 | 0.271 | 0.46 | 0.291 |
| +R | 0.74 | 0.294 | 1.20 | 0.196 | 0.95 | 0.367 | 1.26 | 1.043 | 0.26 | 0.195 |

The amount of photosynthetic pigment is an important physiological parameter concerning photosynthesis efficiency and changes in the content of chloroplast pigments. This can be evidence of stressors tolerance in crops [29–33].

At our experiment, HPM had no reliably changes on the amount of photosynthetic pigments in tomato leaves, but elevated temperatures (30 °C) had increased the amount of photosynthetic pigment (chlorophyll a, chlorophyll b, and carotenoids) compositions in their leaves. Data from other researches showed that, at 15 and 45 min., an exposition time of HPM (10.525 GHz) can increase photosynthetic pigment amounts in wheat seedlings of 7 and 14 days old, whereas the ratios of chlorophyll a/b and carotenoids were higher in seedlings of 7 days old compared with unaffected plants, but they decreased in plants of 14 days old [7]. Scientific studies with electromagnetic field of low-level (900 MHz) exposition on young plantlets of Zea may show different variations in pigments content [29]. Therefore, we can assume that the chlorophyll ratio, provided by irradiated seeds, has

significant variations in comparison with the control value, suggesting HPM sensitivity of photosynthesis efficiency.

Previously, it was observed that the value of the chlorophyll ratio increased slightly with a short exposure time to the electromagnetic field, and a long exposure time decreased the value of the chlorophyll ratio. Moreover, scientists reported an incentive influence of the pulsed electromagnetic field on the accumulation of fresh mass, seedling growth parameters, and seeds germination, as well as on the amount of total chlorophyll (chlorophyll a and b) and dry matter content in maize plants. They determined that electromagnetic waves exposition on seeds can reduce the time period of seed dormancy and improve seed viability together with seedling growth parameters [34]. Experiments with *Brassica napus* L. showed that seed exposure for 30 s with 2.45 GHz frequency microwaves had the most positive effect on seeds viability, plants growth parameters, biomass, photosynthetic pigments content, and activity of antioxidant enzymes (CAT, APX, POD, and SOD) [35].

Previous investigations announced that exposure to HPM had a positive effect on the protein content of 7 days old wheat seedlings and it increased up to 47.6 g kg$^{-1}$ (dry matter) compared with the control, 34.5 g kg$^{-1}$ (dry matter). A reliable increase in the amount of the free amino acid was observed, but the content of total available saccharides decreased compared with the control. Thus, a conclusion was reached where microwaves had a stimulating effect on the content of proteins and free amino acids [7]. The reaction of primary metabolites in our experiment showed that a significantly lower amount of all investigated non-structural carbohydrates (raphinose, sucrose, mannose, fructose, and glucose), which caused a significantly lower total amount of carbohydrates, was detrimental in tomato (+R) leaves. Meanwhile, tomato plants irradiation (+R) at higher temperature (30 °C) conditions had significantly increased content of mannose compared with the control. According to our data, we can confirm that carbohydrates are the energy source for most plant physiological processes such as respiration and cell growth and sugars have important hormone-like functions as primary messengers due to their essential role in plant growth development and metabolic links with primary physiological processes. These changes could be related to carbohydrate contents in plant tissues, affecting the repression of genes and encoding the expression of Rubisco and other photosynthetic proteins under the exposition of HPM radiation and temperature [7,35,36].

Soluble sugars are sources of carbon and energy in cells that play a key role in plant metabolism processes, and their accumulation is permanently regulated as an outcome of the balance amongst supply and carbon use at the entire plant level and for cell sucrose–starch separation, which is controlled by multiple elements, including temperature and microwaves radiation [37]. However, quantitative investigations with physiological and biochemical characteristics disclosed that sugar concentrations before stress were correlated with later stress in plants grown under different conditions [38]. Some external factors (temperature, radiation, etc.) cause a plant's metabolic system reaction—increased efficiency of flavonoids and antioxidant enzymes [39]. It is known that the amount of phenolic acids and flavonoids may fluctuate with respect to various aspects such as cultural practices, environment, and biotic, abiotic, and genetic elements [40].

Significantly higher amounts of phenolic compounds were detected in tomato plants (R) growth from radiated seeds and cultivated at 21 °C. Seed (R) and sprout (+R) irradiation at both (21 °C and 30 °C) plants cultivating temperatures caused significantly higher accumulation of ascorbic acid compared with the control. Meanwhile, plant cultivation at 21 °C and 30 °C had influenced significant lower scavenging of DPPH free-radical activity in tomato plant (R) growth from irradiated seeds and sprouts (+R) compared with the control (Table 4).

Other researchers also reported that total antioxidant activity and total phenolic compound content can be increased by environmental factors such as temperature, radiation, etc. [41,42]. However, plants can create a systemic cell (guard cell) movement reaction, which is mediated by a complicated balance of electric signals related to concentrations of hormone and the activation of the ion channel. Such systemic stomatal reactions likely

evolved to strengthen the suitability of plants and to empower them to acclimatise to environmental changes, assisting in enriching their standing as the main energy converters in sustaining life [24,43]. Our obtained results confirm that the accumulation of secondary metabolites and antioxidants (including several nutritionally important compounds) is a response of the general plant's acclimatization to a variety of environmental factors. Moreover, the experiment elucidated the effects of HPM and evaluated temperature stress in tomato plants based on different HPM exposures (seeds irradiation and sprouts irradiation) to explain the main physiological and antioxidant mechanisms.

**Table 4.** Changes of antioxidant properties of tomato cultivated under combinations of different temperatures and HPM radiation.

| Impact | Total Phenols, mg g$^{-1}$, FW | STDEV | DPPH, µmol g$^{-1}$, FW | STDEV | Ascorbic Acid, mg g$^{-1}$, FW | STDEV |
|---|---|---|---|---|---|---|
| | | | T—21 °C | | | |
| Control | 0.69 | 0.092 | 10.64 | 0.107 | 0.27 | 0.048 |
| R | 0.92 | 0.103 | 10.29 | 0.196 | 1.32 | 0.719 |
| +R | 0.60 | 0.118 | 9.09 | 1.009 | 0.48 | 0.103 |
| | | | T—30 °C | | | |
| Control | 0.59 | 0.173 | 9.00 | 0.208 | 0.23 | 0.065 |
| R | 0.57 | 0.167 | 7.03 | 1.517 | 0.33 | 0.027 |
| +R | 0.49 | 0.139 | 8.51 | 0.231 | 0.37 | 0.064 |

This study showed that there is a great potential for HPM use on biological objects such as seeds and sprouts. It could help to solve some problems related to the biological potential of plants. To obtain more reliable results, further experiments must be carried out with HPM and should be focused on plants productivity and quality attributes. Moreover, it is necessary to admit that there are some limitations according to HPM use on plants related to the device's limited capabilities and plant responses to stressors.

### 4. Conclusions

1.  HPM (9.3 GHz) and elevated temperature exposure upon tomato seeds and sprouts in their primary ontogenetic development level showed a slightly incentive effect on plants growing parameters: dry mass, fresh mass, plants height, and assimilation area.
2.  Elevated temperature (30 °C) increased the amount of photosynthetic pigments (chlorophyll a, chlorophyll b, and carotenoids) in tomato sprout leaves.
3.  Tomato plants (+R) and seeds (R) irradiation significantly reduced the content of non-structural carbohydrates (raphinose, sucrose, mannose, fructose, and glucose) at the temperature of 21 °C.
4.  Seed irradiation by HPM and the elevated temperature caused significantly lower scavenging of DPPH free-radical activity in tomato sprouts.

**Author Contributions:** Conceptualization, A.R. and M.D.; methodology, A.R. and M.D.; software, A.R.; validation, A.R.; S.S. and M.D.; formal analysis, A.R. and S.S.; investigation, A.R., S.S., M.D., R.S., R.K., D.J., R.R. and A.B.; resources, A.R. and M.D.; data curation, A.R. and S.S.; writing—original draft preparation, A.R. and S.S.; writing—review and editing, A.R. and R.K.; visualization, A.R.; supervision, A.R. and M.D.; project administration, M.D.; funding acquisition, A.R. and M.D. All authors have read and agreed to the published version of the manuscript.

**Funding:** This research was funded by a grant MIP-040/2011 from the Research Council of Lithuania.

**Institutional Review Board Statement:** Not applicable.

**Informed Consent Statement:** Not applicable.

**Data Availability Statement:** The data presented in this study are available on request from the authors.

**Acknowledgments:** The manuscript presents research conducted during the long-term LRCAF research program "Horticulture and agro biological bases and technologies".

**Conflicts of Interest:** The authors declare no conflict of interest.

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
