# Peer review of "Differential Physiological Response and Antioxidant Activity Relative to High-Power Micro-Waves Irradiation and Temperature of Tomato Sprouts"

_agriculture, doi:10.3390/agriculture12030422_

Round 1

Reviewer 1 Report

The manuscript requires extensive English editing to improve its readability. The methods used are too simple, and the results do not add additional information to existing literature. Methods are poorly described thus experiments could not be replicated with the information provided. Results are poorly described. Several scientific gaps are shown in the manuscript under consideration, which can not be solved in a revised version. Thus I recommend rejection of the manuscript. 

Author Response

Thank you for you comments and opinion! In the future expirements we will tray to  use more complex methods and more properly describe all experiment.

Reviewer 2 Report

Review report – Agriculture

The manuscript by Audrius Radzevičius et al., entitled " Physiological and antioxidative responses of high power microwaves irradiation and temperature on tomato sprouts" is well written and executed. However, there are some major concerns that need to be resolved before accepting.

Article structure: Authors must follow “authors guidelines” to publish in MDPI journals, a major concern about the article structure, reference style …

*    Please ensure that every reference cited in the text is also present in the reference list (and vice versa). References must follow “Agriculture” style.

Title: The author(s) need to revise the title of the manuscript. The title ‘Physiological and antioxidative responses of on tomato sprouts’ is uncommon in a scientific paper. Hence, I suggest that the title should be “Differential physiological response and antioxidant activity to high-power micro-waves irradiation and temperature of tomato sprouts”.

Abstract: No comments

Keywords: Please add more keywords it will likely attract more readers to your manuscript.

Introduction: The introduction is clearly and well written. Nevertheless, there is still room for improvement.

Materials and Methods: 

  • Please include the reference of each method used in this study.
  • Why drying tomatoes’ sprouts at 105°C, that may alter the chemical composition of the sprouts and, or, volatilize some antioxidant compounds, please include a reference of the used method.
  • Figure 1: The note must be placed after the title using a lowercase character.
  • No information of HPLC column (please specify the column used), validation of the analytical procedures of HPLC is missing, including accuracy, precision, specificity, etc.

Results and Discussion

  • The results section is too short; thus, I suggest combining it with the discussion section to avoid redundancy.
  • Appendix A can be put in the form of a table in the results section.
  • The statistical data is missing. The author(s) should provide extensive statistical analysis in this manuscript, i. e. standard deviation (SD) is missing in Table 1, Table 2, Table 3, and Table 4 as well. While presenting research data, the author(s) should be aware of using adequate statistical measures.

  • Following DPPH (antioxidant assay) results, authors must present in Table 4. either the EC50 the antioxidant concentration that gives the half-maximal response or the IC50 the inhibitory concentration required to reduce the initial concentration of DPPH by 50%.

  • I suggest adding a section to discuss the limitations of the present study.

Author Response

Thank you very much for your valuable comments and opinion about our experiment, we try to do our best to correct all your suggestions.

The manuscript was prepared following by “authors guidelines” and references were corrected. The title changed to “Differential physiological response and antioxidant activity to high-power micro-waves irradiation and temperature of tomato sprouts” and the keywords corrected. The introduction was slightly improved. References to used methods were added

Materials and Methods: 

  • Please include the reference of each method used in this study

References to used methods were added.

  • Why drying tomatoes’ sprouts at 105°C, that may alter the chemical composition of the sprouts and, or, volatilize some antioxidant compounds, please include a reference of the used method.

References to used method were added. Just we can explain that dried sample was intended only for dry matter detection and it could not alter on other our results of chemical composition, because all analysis related with chemical composition were performed in fresh mass.

  • Figure 1: The note must be placed after the title using a lowercase character. Done
  • No information of HPLC column (please specify the column used), validation of the analytical procedures of HPLC is missing, including accuracy, precision, specificity, etc.

Information was clarified and used column specified

 Results and Discussion

  • The results section is too short; thus, I suggest combining it with the discussion section to avoid redundancy.   Results and Discussion section was combined.
  • Appendix A can be put in the form of a table in the results section. Tables put in Results and Discussion section
  • The statistical data is missing. The author(s) should provide extensive statistical analysis in this manuscript, i. e. standard deviation (SD) is missing in Table 1, Table 2, Table 3, and Table 4 as well. While presenting research data, the author(s) should be aware of using adequate statistical measures. New statistical calculations were performed and all the values were expressed as mean ± Standard Deviation (STDV)
  • Following DPPH (antioxidant assay) results, authors must present in Table 4. either the EC50 the antioxidant concentration that gives the half-maximal response or the IC50 the inhibitory concentration required to reduce the initial concentration of DPPH by 50%. Our aim was to detect primary plant response following DPPH (antioxidant assay) results and it is a reliable method to determine the antioxidant capacity of plants. In the future experiments will presents data  according yours valuable observations.

 I suggest adding a section to discuss the limitations of the present study. Added

Thank you very much! Your suggestions were very valuable and it will be very useful in our further experiments.

Reviewer 3 Report

Interesting work. The Discussion section is well done. 
I would like the authors to specify in the methodological part - how many seeds were selected for the experiment? What were the initial indices of seed quality: emergence rate, germination?  It would be desirable to reflect in more detail the description of the measurement technique of leaf assimilation area.

Author Response

Thank you very much for your valuable comments and opinion, we tried to do our best to correct all your suggestions. In section Materials and Methods we indicated number of exposed seeds and their germination, also germination energy. Also description of experiment and performed analysis technics were more precisely described.

Thank you very much!

Reviewer 4 Report

I have read the manuscript, Physiological and antioxidative responses of high power microwaves irradiation and temperature on tomato sprouts.

Keywords must be rephrased.

The topic of the study is interesting and timely. As such, the manuscript could be considered for publication, but not a present form.

References should be written in a format MDPI.

The Introduction is good.

The researchers explained in Materials and Methods: at the first experiment, seeds were exposed using microwaves at 9.3 GHz frequency for 10 minutes, where pulse duration was 4s and pulse repetition – 25 Hz, electric field– 320 kV m-1. At the second experiment, tomato sprouts (14 - days old) were exposed using microwaves at 9.3 GHz frequency for 20 minutes, where pulse duration was 4 s and pulse repetition – 25 Hz, electric field – 320 kV m-1. On what basis were these doses chosen and at this time? A reference must be given. Knowing that the exposure time is very long from my point of view, for example, as it was mentioned in line 238, the exposure time is only 30 seconds.

The temperature on what basis?

The name of tomato variety was explained only. They did not explain the method of cultivation, the number of seeds, the number of treatments, and the area of the experiment. They only explained the cultivation environment?

 The experiment is a laboratory, are there images of it that are valid for display in the MS?

Experimental design,  where on MS ?

Statistical analysis:  Does not mention here the lines 164, 165, 166 and 167, but transfer with experimental design.

leaf area, Assimilating area and assimilation area, what is the difference?

Line 181: unclear phrase ?

R and R+ : mentioned in different meanings in line 18, 19, 174, 188, 193 .

Line 290: What is mean 6 Patents ?

Line 115, 117: at the first experiment, seeds were exposed using microwaves at 9.3 GHz frequency for 10 minutes, where pulse duration was 4 s and pulse repetition – 25 Hz, electric field– 320 kV m-1. At the second experiment, tomato sprouts (14 - days old) were exposed using microwaves at 9.3 GHz frequency for 20 minutes, where pulse duration was 4 s and pulse repetition – 25 Hz, electric field – 320 kV m-1. How Two experiments ?

It is useful to display data in [Figures. Use Duncan instead of test LSD test and provide deviations for all given measurements (SE or SD)

The discussion part must talk about your data and clarify the reasons for your results, not to repeat previous studies. Please, improve the discussion part.

Thanks.

Author Response

Thank you very much for your valuable comments and opinion related with our experiment, we tried to do our best to correct all your suggestions.

Please, see our answers:

Keywords must be rephrased. Keywords were corrected.

References should be written in a format MDPI. The manuscript was corrected according “Authors guidline”

The Introduction is good. Thank you!

The researchers explained in Materials and Methods: at the first experiment, seeds were exposed using microwaves at 9.3 GHz frequency for 10 minutes, where pulse duration was 4s and pulse repetition – 25 Hz, electric field– 320 kV m-1. At the second experiment, tomato sprouts (14 - days old) were exposed using microwaves at 9.3 GHz frequency for 20 minutes, where pulse duration was 4 s and pulse repetition – 25 Hz, electric field – 320 kV m-1. On what basis were these doses chosen and at this time? A reference must be given. Knowing that the exposure time is very long from my point of view, for example, as it was mentioned in line 238, the exposure time is only 30 seconds. Reference was given in the text and it came from our previous experiments with HPM. So, the irradiation time and frequency were selected according to the results of our previous studies. We would like to admit that most studies were made with low frequency microwaves and without elimination of heating effect (so irridiation can not be exposed in longer time period).  In our experiment with HPM 9.3 GHz frequency we used inpulses and avoided heat effect and it is made possible to extend irradiation time.

The temperature on what basis? Temperature, also, were selected according to previous studies with tomato plants.

The name of tomato variety was explained only. They did not explain the method of cultivation, the number of seeds, the number of treatments, and the area of the experiment. They only explained the cultivation environment? In Materials and Methods section more accurate explanation and used methods were added.

Experimental design,  where on MS ? “Experimental design” added to Materials and Methods section.

Statistical analysis:  Does not mention here the lines 164, 165, 166 and 167, but transfer with experimental design.  Corrected.

leaf area, Assimilating area and assimilation area, what is the difference? It has no difference, everywhere we had change into “assimilation area”

Line 181: unclear phrase ? Corrected.

R and R+ : mentioned in different meanings in line 18, 19, 174, 188, 193 . Corrected.

Line 290: What is mean 6 Patents ? Corrected – it was  a technical mistake.

Line 115, 117: at the first experiment, seeds were exposed using microwaves at 9.3 GHz frequency for 10 minutes, where pulse duration was 4 s and pulse repetition – 25 Hz, electric field– 320 kV m-1. At the second experiment, tomato sprouts (14 - days old) were exposed using microwaves at 9.3 GHz frequency for 20 minutes, where pulse duration was 4 s and pulse repetition – 25 Hz, electric field – 320 kV m-1. How Two experiments ? It is really one common experiment, just it was divided in two parts to better understand how all experiment was carried out.

It is useful to display data in [Figures. Use Duncan instead of test LSD test and provide deviations for all given measurements (SE or SD). New statistical calculations were performed and all the values were expressed as mean ± Standard Deviation (STDV)

The discussion part must talk about your data and clarify the reasons for your results, not to repeat previous studies. Please, improve the discussion part. Results and Discussion section was combined and improved.

Thank you very much! Your suggestions were very valuable and it will be very useful in our further experiments.

Round 2

Reviewer 1 Report

The authors properly attended to the comments from this reviewer. I recommend acceptance in its present form. 

Reviewer 2 Report

Because the manuscript has been revised well according to my suggestions

 I think it can be accepted now.

Reviewer 4 Report

Authors have addressed all of my queries. Revised version is significantly improved and can be accepted for publication.